🔓 | Open Peer Review | Clinical Microbiology | Research Article

# Quantitating SARS-CoV-2 neutralizing antibodies from human dried blood spots

Katherine Berman,[1] Greta Van Slyke,[1] Hayley Novak,[1] Jean M. Rock,[1] Rachel Bievenue,[1] Amanda K. Damjanovic,[1] Kate L. DeRosa,[1] Gianna Mirabile,[1] Kara Phipps,[1] Jessica Machowski,[1] Sean Bialosuknia,[1] Roxie C. Giradin,[1] Alan P. Dupuis,[1] Anne F. Payne,[1] William T. Lee,[1] Kathleen A. McDonough,[1] Monica M. Parker,[1] Linda M. Styer,[1] Nicholas J. Mantis[1]

**ABSTRACT** In the earliest days of the COVID-19 pandemic, the collection of dried blood spots (DBS) enabled public health laboratories to undertake population-scale seroprevalence studies used to estimate rates of SARS-CoV-2 exposure. With SARS-CoV-2 seropositivity levels now estimated to exceed 94% in the United States, attention has turned to using DBS to assess neutralizing antibodies within cohorts of interest. With this goal in mind, we generated contrived DBS (cDBS) and whole blood-derived DBS from convalescent and vaccinated individuals and subjected DBS eluates to a battery of assays, including a SARS-CoV-2 multiplexed microsphere immunoassay (MIA), a receptor binding domain (RBD)-human ACE2 inhibition assay (iACE2), a cell-based pseudovirus neutralization assay, and real-time PCR-based surrogate neutralization assay (NAB-Sure). The DBS results were benchmarked against paired serum samples tested in a clinically validated SARS-CoV-2 plaque reduction neutralization titer (PRNT) assay. The results of an 8-plex MIA and NAB-Sure assays demonstrated highly significant correlations with PRNT values when evaluated with a panel of 86 paired serum–DBS samples. Both the MIA and NAB-Sure are adaptable to automated liquid handlers for high-throughput capacity. While neutralizing assays were limited to the ancestral SARS-CoV-2 WA1, this study nonetheless represents an important proof of concept demonstrating the potential utility of DBS as a biospecimen type for use in assessing immunity to SARS-CoV-2 at the community and population levels.

**IMPORTANCE** SARS-CoV-2 variants of concern continue to circulate globally and remain a serious health threat to large segments of the population. From a public health standpoint, identifying vulnerable communities based on immune status is critical in terms of vaccine booster recommendations. In this report, we investigated the utility of dried blood spots (DBS) as a biospecimen type from which to estimate SARS-CoV-2 neutralizing antibody titers. Using contrived and whole blood-derived DBS, we demonstrate that SARS-CoV-2 neutralizing antibodies are readily measurable in DBS eluates and correlate with plaque reduction neutralization titer (PRNT) values from paired serum samples. Moreover, several of the methods used to estimate SARS-CoV-2 neutralizing antibodies in DBS eluates are adaptable to high-throughput platforms.

**KEYWORDS** serology, neutralizing, COVID-19, antibody, human

The collection and serologic analysis of dried blood spots (DBS) from population-based serosurveys proved integral to understanding the dynamics of SARS-CoV-2 infection in the first months of the COVID-19 pandemic. In New York State, for example, a statewide seroprevalence study among ~15,000 individuals in public settings (e.g., grocery store fronts) conducted in April 2020 afforded some of the first insights into the overall disease incidence and demographics of virus exposure in urban and

**Peer Reviewer** Elias A. Rahal, American University of Beirut, Beirut, Lebanon

Address correspondence to Linda M. Styer, Linda.Styer@health.ny.gov, or Nicholas J. Mantis, Nicholas.Mantis@health.ny.gov.

The authors declare no conflict of interest.

See the funding table on p. 15.

rural communities (1). Similarly, a cross-sectional serosurvey of ~400,000 newborns in New York State that employed heel stick DBS collected between November 2019 and November 2021 afforded unprecedented insights into COVID-19 exposures and vaccination status among a diverse population of females who recently gave birth (2). Other groups have leveraged clinician-assisted and self-collected DBS to conduct large-scale serosurveys of vulnerable cohorts, school age children, and otherwise hard-to-reach communities (3–11). In addition to ease of collection, DBS samples are stable on a Whatman filter paper for long periods of time under refrigeration (4°C–8°C) or frozen (up to −20 °C) in the presence of desiccants (12).

During the COVID-19 pandemic, DBS have largely been employed in serological studies aimed at determining SARS-CoV-2 spike (S) and nucleocapsid (N) reactivity as measures of infection and vaccination. However, with SARS-CoV-2 seropositivity levels now estimated to exceed 94% in the United States, the value of such serosurveys has diminished (13). Furthermore, the emergence of Omicron subvariants with highly mutated spike proteins confounds the ability to extrapolate neutralizing antibody titers from binding antibody units (BAUs) (14, 15). For those and other reasons, there is considerable interest in developing a method to assess SARS-CoV-2 neutralizing titers directly from DBS eluates.

Indeed, several reports have already described the use of DBS eluates to assess the functional activity of SARS-CoV-2 antibodies. For example, Itell and colleagues examined paired plasma and fingerstick (FS) DBS from convalescent individuals in SARS-CoV-2 spike-pseudotyped lentiviral particle assay in 96- and 384-well formats (16). In another report, Sancilio and colleagues used an ACE2-RBD binding assay on the Mesoscale Diagnostics (MSD) platform to estimate neutralizing titers in contrived DBS (cDBS) eluates (17) and in self-collected DBS from a community-based serology survey (18). However, neither study benchmarked their results against plaque reduction neutralization titer (PRNT) values, which is considered the gold standard for determining SARS-CoV-2 neutralizing antibody levels in vaccinated and convalescent individuals (19). Therefore, to advance the utility and validity of DBS for the purpose of functional SARS-CoV-2 serology, we investigated the compatibility of contrived DBS eluates with different SARS-CoV-2 neutralization assays and compared their outputs to PRNT values derived from matched serum samples.

## RESULTS

Performing PRNT assays on the thousands of DBS generated as part of population-scale serosurveys for COVID-19 is neither practical nor technically feasible (2, 8, 9). Indeed, in pilot studies, DBS eluates proved to be incompatible with PRNT. Hence, we sought to investigate alternative (proxy) assays as reliable measures of SARS-CoV-2 neutralizing antibody titers in DBS. To achieve this goal, we initially focused on the use of contrived DBS (cDBS) samples from three commercially available serum panels: Panel D consists of individuals with a defined history of SARS-CoV-2 infection prior to June 2020, Panel E consists of serum samples from a pre-pandemic cohort, and Panel H consists of a fully COVID-19 vaccinated cohort. Contrived DBS were produced from each by mixing an equal volume of serum with fresh human blood cells (HBCs) and then spotting (50 μL) onto Whatman 903 filter paper (Fig. 1). DBS cards were allowed to dry at room temperature for ~4 hours and then stored until needed at −20°C with desiccant. For collection of DBS eluates, 3- or 6-mm punches are excised from the Whatman filter paper card and incubated in an assay-specific elution buffer (Fig. 1).

### An 8-plex microsphere immunoassay (MIA) for assessment of SARS-CoV-2 S- and N-specific IgG in human DBS eluates

It is recognized that spike antibody titers correlate strongly with SARS-CoV-2 neutralizing antibody levels (14, 20). To discern whether this also applies to DBS eluates, we first optimized and validated a SARS-CoV-2 8-plex MIA, which includes WA1 spike full-length trimer (FLT), spike full-length monomer (FLS), S1, RBD, and three different N antigens

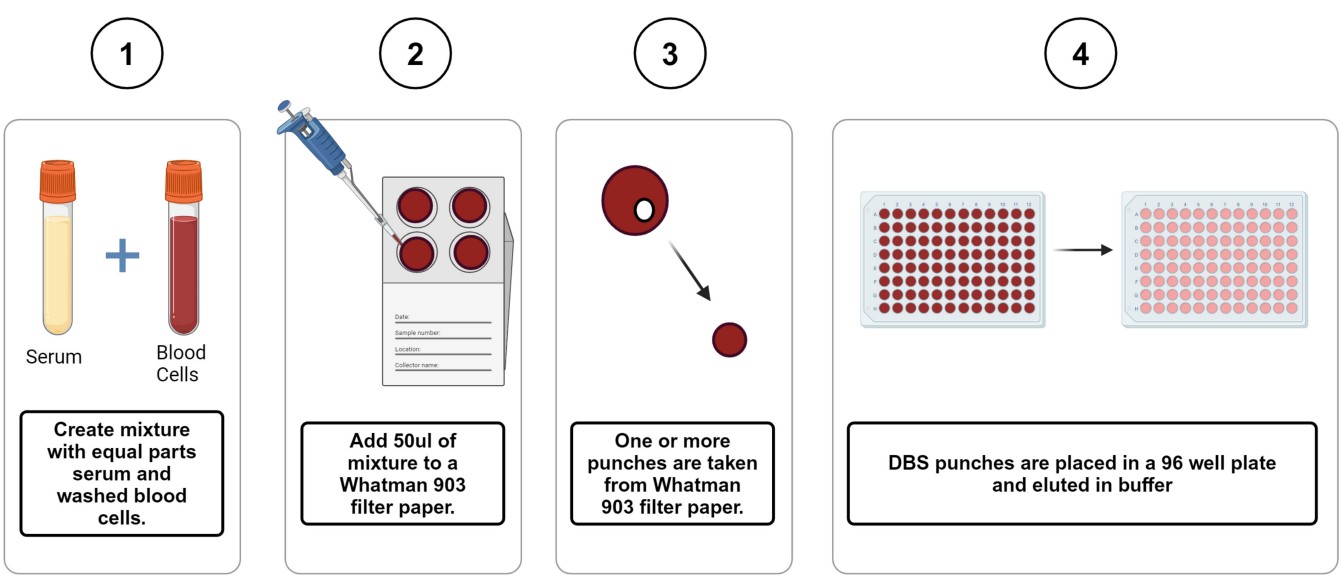

**Generation of Contrived DBS Eluates**

**1** Serum + Blood Cells

**Create mixture with equal parts serum and washed blood cells.**

**2**

**Add 50ul of mixture to a Whatman 903 filter paper.**

**3**

**One or more punches are taken from Whatman 903 filter paper.**

**4**

**DBS punches are placed in a 96 well plate and eluted in buffer**

**FIG 1** Generation of contrived DBS (cDBS) eluates. (A) Schematic showing the generation of contrived DBS (cDBS), DBS punches, and eluate collection.

(Table S1). To elute DBS, a single 3-mm punch is excised from the card and incubated in 250 µL Tris-buffered saline with 1% casein for 1 hour at room temperature without shaking. This elution condition was determined to be optimal in terms of SARS CoV-2 antibody recovery (Fig. S1). The 8-plex MIA affords improved sensitivity and specificity over the 2-plex MIA used previously for large-scale serosurveys (8, 21). Notably, the sensitivity of the 8-plex MIA was assessed in two ways. First, SARS-CoV-2 antibody-positive plasma samples ($n$ = 3) were twofold serially diluted and tested on the FDA-approved AdviseDx SARS-CoV-2 IgG II (Abbott). In parallel, plasma dilutions were combined with equal parts HBC and spotted onto DBS cards to create paired cDBS that were tested with the 8-plex MIA. MIA reactivity levels in cDBS eluates had similar reactivity profiles to the AdviseDx (Table S2). The 8-plex MIA was also assessed using a commercially available seroconversion panel. Antibody reactivity with FLS and FLT was positive at the 36-day point compared to the S1 antigen (and RBD), which was not positive until day 50 (Table S3). The S1 antigen was the only spike-derived antigen employed in the previous 2-plex MIA (21).

In addition to the seven SARS-CoV-2 antigens, the 8-plex MIA includes an internal control bead to verify the addition of primary (i.e., plasma/serum/eluate) and secondary (human anti-IgG-PE) reagents. While tetanus toxoid (TT) is commonly used as an internal control (16), we found that TT reactivity was too variable in pediatric and adult populations. Instead, among the 32 different anti-IgG antibodies and antigens tested, we selected an anti-human IgG3 capture antibody as a control because it produced consistently positive results among pediatric and adult DBS samples that were within the dynamic range of our Luminex instrument and sensitive to changes in the amount of sample added to the assay (Fig. S2).

## PRNT values correlate with SARS-CoV-2 8-plex MIA

Using Panel D paired serum and cDBS samples, we investigated the degree to which 8-plex MIA values correlated with PRNT values. SARS-CoV-2 Washington (WA1) neutralizing antibody levels were determined using a clinically validated PRNT assay at the

Wadsworth Center (19, 22, 23). The characteristics of the Panel D cohort are presented in Table 1. The average $PRNT_{50}$ for the cohort was 351 (+/− 271). The percent SARS-CoV-2 neutralization at a fixed 1:100 serum dilution (average 55 +/− 22%) was extrapolated based on PRNT curves to enable direct comparison with values derived from DBS eluates (see below). In parallel, eluates from Panel D and Panel E cDBS were generated, as shown in Fig. 1, and analyzed using the optimized 8-plex MIA. SARS-CoV-2 antigen-specific MFI values in cDBS eluates were significantly greater than Panel E cDBS eluates (Fig. S3), confirming the specificity of the 8-plex MIA for COVID-19-positive individuals. Moreover, there were highly significant correlations between Panel D spike-specific MFI values (FLT, FLS, S1, and RBD) and PRNT, with Pearson r values between 0.76 and 0.83 (Fig. 2). Summing the MFI values of the four different spike antigens also resulted in a significant correlation with PRNT (r = 0.81). In contrast, the relationship between nucleocapsid reactivity and PRNT was not significant. We conclude that the MFI values derived from DBS eluates to a given SARS-CoV-2 spike antigen correlate with the virus neutralizing activity, as defined by PRNT.

## Pseudovirus neutralization activity in cDBS eluates

Non-replicating, lentiviral-based pseudoviruses expressing SARS-CoV-2 trimeric spike proteins are employed widely to assess SARS-CoV-2 neutralizing antibody titers in plasma and serum (24–27). However, it is unclear whether these same assays are compatible with DBS eluates in terms of yielding SARS-CoV-2 neutralizing antibody titers in line with PRNT. To investigate this, we employed a commercially available non-replicating reporter virus particles (RVP) expressing the SARS-CoV-2 spike protein (D648G) and encoding *Renilla* luciferase (Fig. 3) (28). We confirmed RVP efficacy using the human monoclonal antibody CC12.3, which neutralizes SARS-CoV-2 with a reported $IC_{50}$ of 0.026 µg/mL (27). In our study, CC12.3 had an $IC_{50}$ of 0.04 µg/mL (Fig. 3A). Moreover, the 15 Panel D serum samples had neutralization activities that ranged from 15% to 98% with high concordance with PRNT (r = 0.67; *P* = 0.007) (Fig. 3B). Thus, RVP serves as a reliable measure of SARS-CoV-2 activity in the context of serum.

**TABLE 1**   Summary of Panel D cohort and test results[h]

| # | Age | Sex | PRNT (serum) | | MFI in 8-plex MIA (DBS) | | | | | | iACE2 (DBS) | RVP (DBS) | | NAB (DBS) |
|---|---|---|---|---|---|---|---|---|---|---|---|---|---|---|
| | | | $NT_{50}$ | %Neut | N[a] | S1[b] | FLS[c] | FLT[d] | RBD[e] | Sum[f] | iACE2[g] | NT50 | %Neut | $NT_{50}$ |
| 1 | 38 | F | >832 | 93 | 6121 | 4915 | 10178 | 14238 | 11185 | 40516 | 43 +/− 5.3 | 379 | 89 | 712 |
| 2 | 45 | F | 104 | 51 | 3283 | 521 | 1514 | 2557 | 1509 | 6100 | 17 +/− 4.3 | 183 | 65 | 90 |
| 3 | 44 | F | 208 | 42 | 2925 | 2377 | 4880 | 6843 | 5306 | 19406 | 22 +/− 2.2 | 151 | 58 | 244 |
| 4 | 48 | F | >832 | 93 | 3396 | 4295 | 9836 | 15808 | 11335 | 41274 | 53 +/− 2.7 | 203 | 79 | 557 |
| 5 | 55 | F | 416 | 33 | 544 | 872 | 2779 | 4562 | 2868 | 11081 | 14 +/− 3.4 | 435 | 67 | 196 |
| 6 | 65 | F | 208 | 60 | 4560 | 3221 | 6926 | 7929 | 5972 | 24047 | 14 +/− 3.4 | 395 | 64 | 270 |
| 7 | 40 | F | 208 | 77 | 7424 | 1714 | 4967 | 6497 | 5496 | 18674 | 18 +/− 2.3 | 513 | 67 | 189 |
| 8 | 39 | M | 832 | 70 | 6465 | 4561 | 8677 | 16991 | 10578 | 40806 | 42 +/− 7.1 | 262 | 71 | 899 |
| 9 | 34 | F | IND | 30 | 1434 | 1147 | 2983 | 3948 | 2873 | 10951 | 17 +/− 1.8 | 239 | 68 | 249 |
| 10 | 46 | F | 208 | 40 | 11565 | 433 | 2941 | 9150 | 2051 | 14575 | 15 +/− 4.3 | 229 | 68 | 176 |
| 11 | 30 | M | 208 | 47 | 6989 | 3387 | 7843 | 9969 | 6513 | 27711 | 16 +/− 3.4 | 141 | 64 | 356 |
| 12 | 47 | F | 208 | 40 | 2468 | 1432 | 3145 | 5014 | 4503 | 14093 | 17 +/− 4.2 | 188 | 65 | 249 |
| 13 | 57 | F | 26 | 21 | 1455 | 736 | 1907 | 2948 | 2176 | 7766 | 12 +/− 1.2 | 280 | 68 | 114 |
| 14 | 36 | M | 416 | 60 | 5371 | 2682 | 6136 | 9272 | 5869 | 23958 | 24 +/− 5.1 | 209 | 62 | 364 |
| 15 | 30 | F | 208 | 72 | 7691 | 2296 | 6687 | 9218 | 5867 | 24068 | 21 +/− 2.9 | 290 | 67 | 321 |

[a]N, SARS-CoV-2 nucleocapsid.
[b]S1, SARS-CoV-2 spike subunit 1.
[c]FLS, SARS-CoV-2 full-length spike.
[d]FLT, SARS-CoV-2 full-length trimer.
[e]RBD, SARS-CoV-2 receptor-binding domain.
[f]Sum, represents the total of all spike antigens.
[g]iACE2 indicates inhibition of biotinylated ACE2 binding to RBD or FLT antigen complexed to MagPlex beads.
[h]MIA = microsphere immunoassay; MFI = median fluorescence intensity; NAB = NAB-Sure neutralizing antibody; RVP = reporter virus assay.

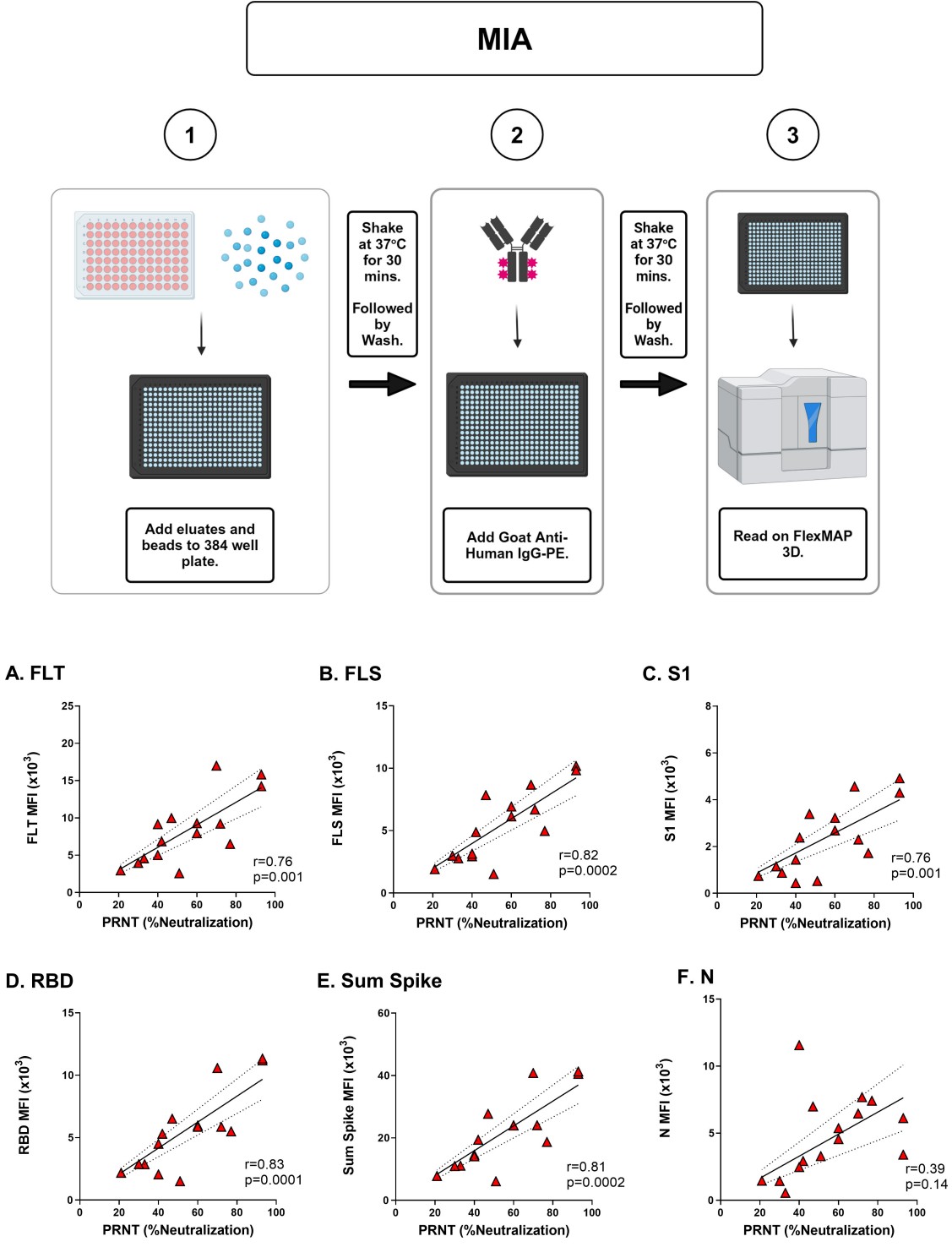

**FIG 2** Correlation between PRNT and MIA with cDBS eluates. (Top) Workflow for the 8-plex MIA. (Bottom) Subset of Panel D cDBS eluates ($n$ = 15) underwent testing via 8-plex MIA and were compared to paired serum samples tested by PRNT. Pearson correlation coefficient of MFI values and SARS-CoV-2 neutralizing activity (%) for the following antigens: (A) FLT; (B) FLS; (C) S1; (D) RBD; and (F) N. The Pearson r values and $P$-values are shown as insets within each panel.

To determine if the RVP assay is compatible with cDBS, we collected eluates from three 3 mm punches from 15 Panel D and 7 Panel E cDBS. DBS eluates were then diluted 1:50 into DMEM-HEPES medium and then mixed with RVPs prior to application to 293T cells constitutively expressing human ACE2 (see **Materials and Methods**). The Panel D eluates had 60%–90% neutralization activity [average 68%], but the correlation with

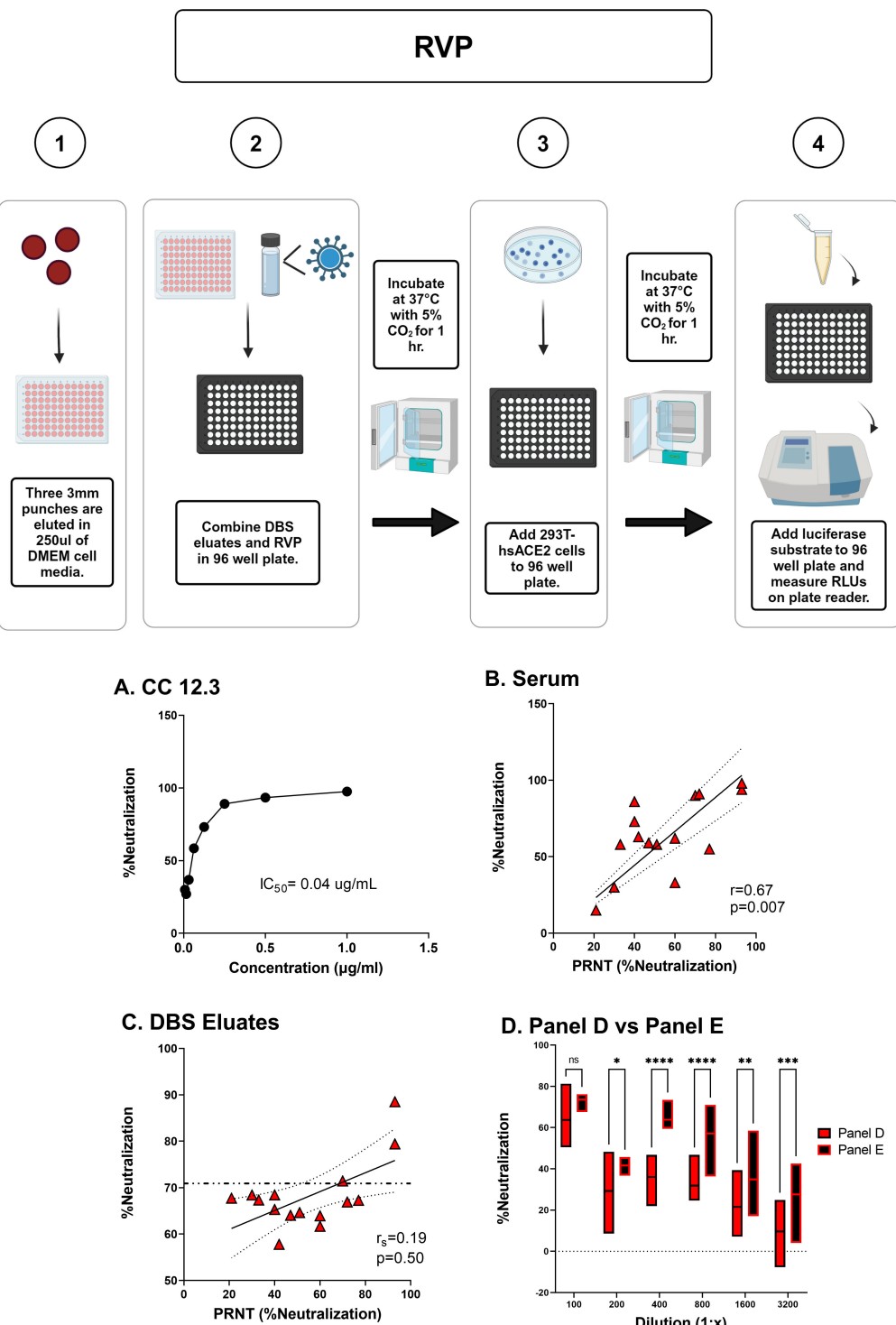

**FIG 3** Compatibility of cDBS eluates with the RVP neutralizing assay. (Top) Workflow for the RVP neutralizing assay. (Bottom) (A) Neutralizing titers associated with CC 12.3 with $IC_{50}$ shown as the text. (B) Correlation between PRNT (% neutralization) and RVP (% neutralization) with matched Panel D serum samples. (C) Correlation between PRNT (% neutralization) generated with Panel D serum versus RVP generated (% neutralization) from matched cDBS eluates. (D) Comparison of RVP neutralization of Panel D ($n = 15$) and Panel E ($n = 7$) cDBS eluates during serial dilutions. Asterisks indicate a significant difference between groups by two-way ANOVA, where *$P = 0.02$, **$P = 0.009$, ***$P = 0.0002$, and ****$P < 0.0001$. For (B), Pearson r and $P$-value are shown as a panel inset. For (C), Spearman $r_S$ and $P$-value are shown as a panel inset. DBS eluates were generated by incubating the punches in DMEM-HEPES at room temperature for 2 hours with agitation.

PRNT was not significant ($P$ = 0.50) (Fig. 3C). Moreover, Panel E eluates (negative controls) had "neutralization" activity that was essentially equivalent to that of Panel D (Fig. 3D), demonstrating that the DBS eluates alone were interfering with the RVP assay. Efforts to improve the sensitivity of the assay across a range of dilutions, alternative elution buffers, and removal of hemoglobin using reagents designed for this purpose (e.g., Hemoglo-Bind) were unsuccessful at reducing the background activity (Fig. 3D; data not shown). Thus, we conclude that cDBS eluates from convalescent (unvaccinated) individuals have insufficient virus neutralizing titers to distinguish them from unvaccinated controls in the pseudovirus (RVP) platform.

## Relationship between PRNT and iACE2

We next evaluated cDBS eluates for the ability to inhibit human ACE2–RBD interactions as a surrogate measure for SARS-CoV-2 neutralizing antibody titers. In the Luminex-based assay (iACE2), RBD- and FLT-coupled microspheres were incubated with soluble biotinylated-ACE2 in the absence or presence of competitor antibodies (e.g., monoclonal, antisera, or DBS eluates) and then probed with streptavidin–PE and analyzed by using FlexMap 3D (Fig. 4). A reduction in the MFI was interpreted as antibody-mediated inhibition of ACE2 binding to RBD- and FLT. The iACE2 assay was verified using the well-characterized monoclonal antibody CC12.3 (27). In the iACE2 assay, CC12.3 had an $IC_{50}$ value for WT FLT and RBD of 0.04 ug/mL and 0.02 ug/mL, respectively (Fig. S4). These values are consistent with those reported by Rogers and colleagues (27).

We next examined Panel D DBS eluates in the iACE2 assay, with the expectation that Panel D eluates would have greater inhibition activity than Panel E eluates. Indeed, Panel D had significantly higher iACE2 values (23% average; range 10%–50%) as compared to Panel E (9% average; range 0%–15%) (Fig. 4). In terms of the correlation with PRNT, Panel D iACE2 values had an $r_s$-value of 0.71 ($P$ = 0.004) for RBD-coated beads and an r-value of 0.72 ($P$ = 0.003) for FLT-coated beads. iACE2 (%) values for FLT-coated beads ranged from 25% to 40% (avg 31%) (Fig. 4; Fig. S4). Panel D DBS eluates performed slightly less well than the matched serum samples from which they were derived (Fig. S4), indicating that a degree of ACE2-RBD inhibitory activity is lost during the collection of cDBS eluates. Nonetheless, these results demonstrate that the iACE2 assay affords a reliable estimate of SARS-CoV-2 neutralization potential using cDBS eluates.

## Determining SARS-CoV-2 neutralizing activity in DBS eluates using NAB-Sure

NAB-Sure is a commercially available cell-free assay based on the successive proximity extension amplification reaction (SPEAR) technology with qPCR as a readout (SpearBio, Woburn, MA). Disruption of a proximity extension amplification reaction primed by recombinant SARS-CoV-2 S1 protein and human ACE2 by neutralizing antibodies in DBS eluates results in increased CT values. To investigate this technology, we first evaluated the NAB-Sure assay using a set of 11 previously authenticated DBS eluates provided by the manufacturer and which were subjected to five threefold serial dilutions (2 to 486) and tested in duplicate using in-house instrumentation. The NAB-Sure assay revealed $NT_{50}$ values that ranged from 210 to 9,500 (avg. 2,333) with 17.7% CV (Fig. 5A). There was a significant correlation between $NT_{50}$ values from DBS eluates tested by NAB-Sure and paired serum samples tested by PRNT [Spearman's $r_s$ = 0.97; $P$ < 0.0001] (Fig. 5A). Thus, NAB-Sure was compatible with DBS eluates and had a high concordance with PRNT.

We then used the NAB-Sure assay to evaluate cDBS eluates from Panel D and E. Samples were tested three times in independent reactions under identical assay conditions. The variability between runs was minimal as the average %CV across samples was 20.3%. Overall, average NAB-Sure inhibition values were significantly higher in the Panel D eluates as compared to Panel E (Table S4). Moreover, Panel D samples had $NT_{50}$ values that correlated with PRNT results ($r_s$ = 0.75; $P$-value = 0.009) (Fig. 5B), indicating that NAB-Sure is sufficiently sensitive to detect SARS-CoV-2 neutralizing titers in cDBS eluates derived from naturally infected individuals from the earliest phases of the COVID-19 pandemic.

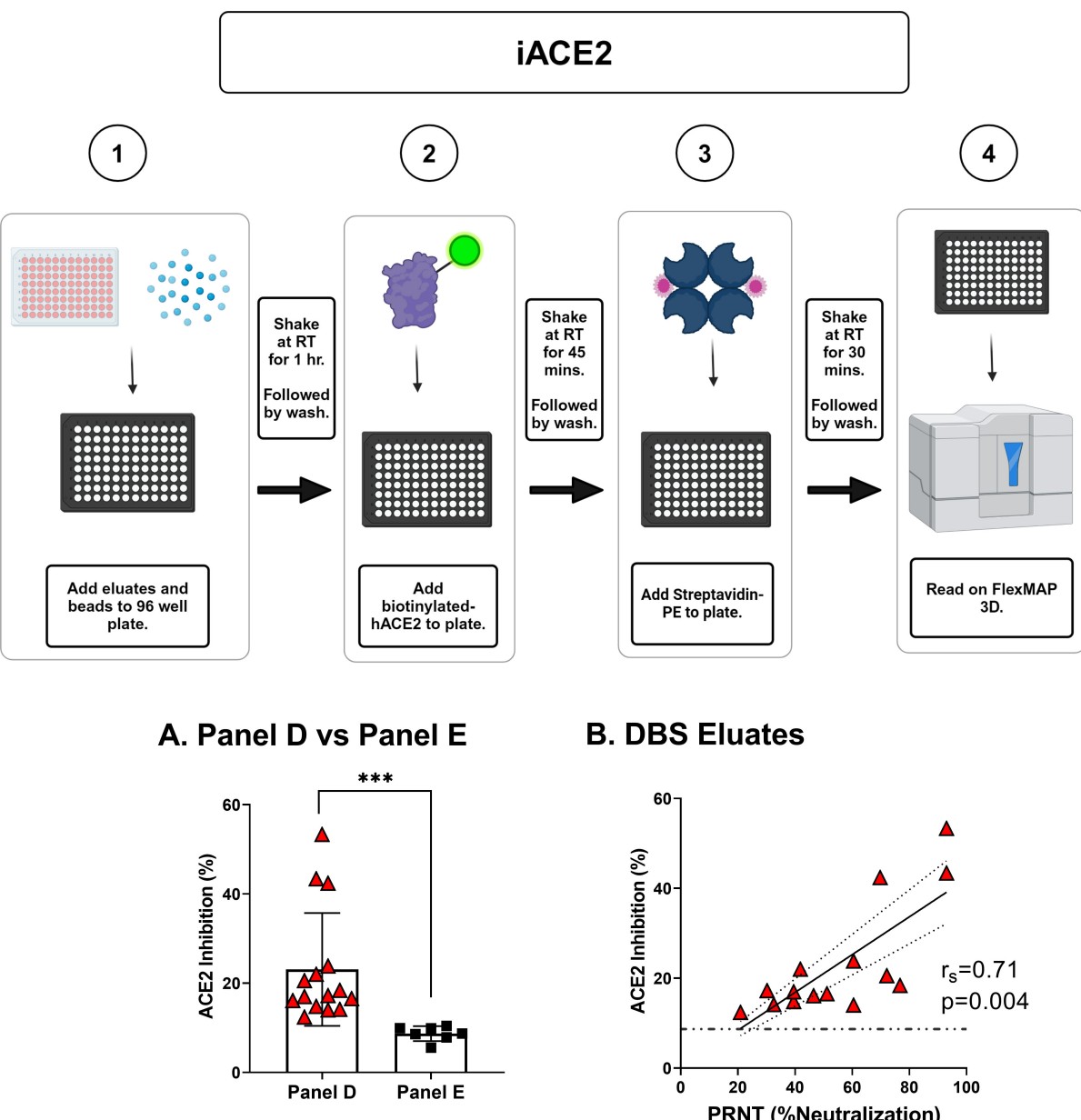

**FIG 4** Correlation between Luminex-based ACE2 inhibition and SARS-CoV-2 neutralization activity. (Top) Workflow for the ACE2 inhibition (iACE2) assay. (A) Examination of the iACE2 activity in Panel D cDBS eluates ($n = 15$) versus Panel E eluates ($n = 7$). In (A), asterisks indicate a significant difference between groups by Welch's t-test, where ***$P = 0.0006$. (B) Correlation of iACE2 values from Panel D cDBS eluates, as shown in (A), to SARS-CoV-2 neutralizing activity (%) from paired serum samples analyzed via PRNT. cDBS eluates were evaluated in triplicate; shown are average values. The Spearman $r_s$ and P-value are shown as a panel inset for (B). The dotted line on (B) represents the average iACE2 activity for the SARS-CoV-2 negative cohort (Panel E) and indicates the threshold for positive responses.

## Assessing SARS-CoV-2 neutralizing activity in cDBS eluates from vaccinated individuals

Studies prior to this point relied on DBS eluates derived from SARS-CoV-2 infected individuals with relatively low overall spike-specific antibody titers and low SARS-CoV-2 neutralizing levels. With an estimated >90% of the United States population being seropositive for SARS-CoV-2 due to vaccination and/or natural infection at this point in time (13), we wished to revisit the utility of DBS eluates in assessing virus-specific antibody titers and neutralizing activity in a vaccinated cohort. For this purpose, we generated cDBS from a commercially available panel (Panel H) of serum samples

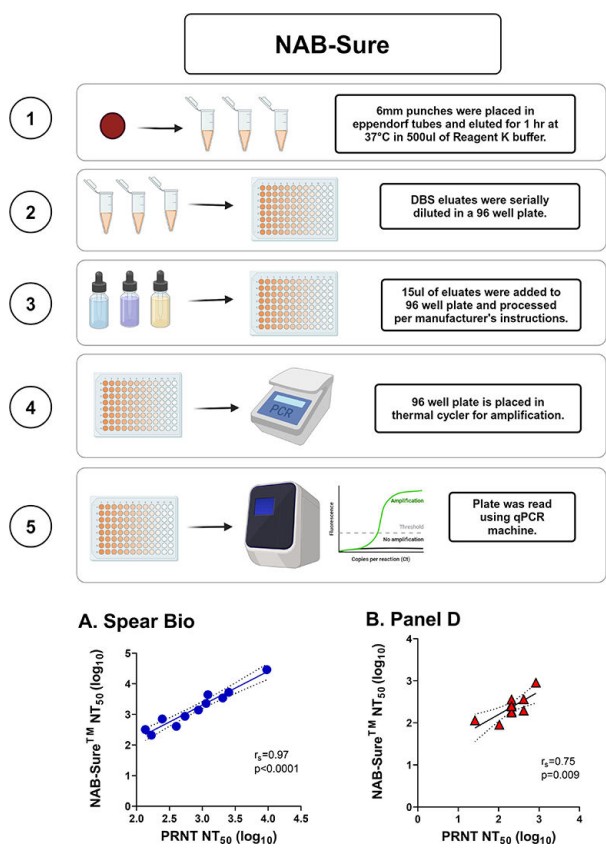

**FIG 5** NAB-Sure exhibits high concordance with PRNT. (Top) Workflow for the NAB-Sure assay. (A) Spearman's correlation of NAB-Sure $NT_{50}$ ($\log_{10}$) from authentic DBS sample eluates ($n = 11$) compared to PRNT paired serum. DBS samples were evaluated in duplicate; shown are average values. (B) Spearman correlation comparing NAB-Sure $NT_{50}$ ($\log_{10}$) for cDBS eluates and paired serum $NT_{50}$ ($\log_{10}$) from PRNT for 14 Panel D samples performed in triplicate; shown are average values. For (A) and (B), Spearman's $r_s$ and $P$ values are shown as panel insets.

collected from individuals who had received two doses of the original Moderna (mRNA 1273) vaccine. Panel H DBS eluates ($n = 14$) had average RBD-specific antibody levels of 15,746 ± 1,790 MFI compared to 10,392 ± 3,325 MFI for Panel D and 351 ± 313 MFI for Panel E (Fig. 6A).

Panel H DBS eluates were next evaluated using the RVP (pseudovirus) assay (Fig. 6B), the iACE2 assay (Fig. 6C), and NAB-Sure (Fig. 6D). The results demonstrated that all three platforms (iACE2, RVP, and NAB-Sure) were compatible with DBS eluates, as evidenced by Panel H samples exhibiting RVP neutralization (%), which was statistically higher than that of Panel D and Panel E samples. Panel H samples also maintained high RVP neutralization (%) even at dilutions as high as 1:1,600 (Fig. 6B). When evaluated using the iACE2 assay, the average ACE2 inhibition was at 60%, with some samples approaching 100% (Fig. 6C). When assessed using NAB-Sure, $NT_{50}$ values approached 10,000 (Fig. 6D). Thus, all three platforms (iACE2, RVP, and NAB-Sure) are amenable to cDBS eluates derived from COVID-19 vaccinated individuals.

## Assessing SARS-CoV-2 neutralizing activity using DBS samples from a clinical cohort

Until this point, our analysis involved a rather small number of cDBS samples derived from commercial panels. To extend our findings to primary samples, we generated dried blood spots ($n = 86$) from available ethylenediaminetetraacetic acid (EDTA) whole blood drawn from participants with a range of infection and vaccination histories and with

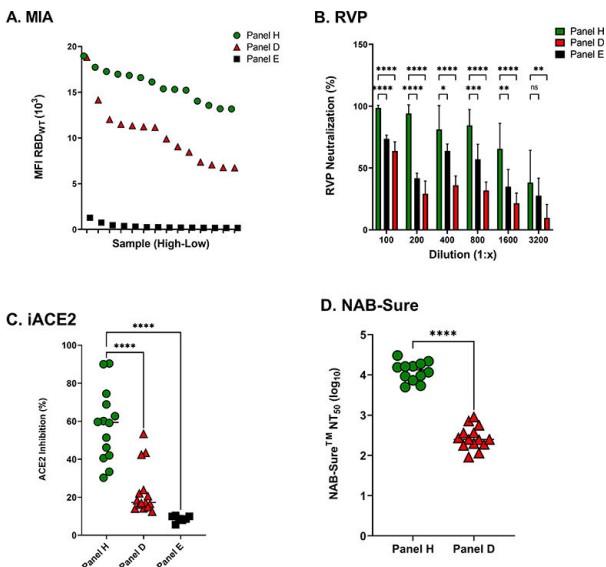

**FIG 6** SARS-CoV-2 neutralizing titers in cDBS eluates from vaccinated individuals. (A) cDBS eluates from Panels H, D, and E were rank ordered from high to low (left to right) based on the $RBD_{WT}$ MFI determined in the 8-plex MIA. (B) Comparison of RVP neutralization % of Panels H, D, and E during serial dilutions. (C) Comparison of the iACE2 activity in Panels H, D, and E. (D) Comparison of $NT_{50}$ ($log_{10}$) from NAB-Sure between Panel H and Panel D. Two of the highest Panel H eluates were diluted 1:2 prior to the NAB-Sure assay to achieve $NT_{50}$ ($log_{10}$) values. For (B), asterisks indicate a significant difference between groups by two-way ANOVA, where *$P = 0.02$, **$P = 0.009$, ***$P = 0.0002$, and ****$P < 0.0001$. For (C) and (D), asterisks indicate a significant difference between groups using Welch's $t$-test, where ****$P < 0.0001$.

$PRNT_{50}$ values derived from paired serum collected at the same time as the whole blood. DBS eluates were subjected to RBD-specific MIA (Fig. 7A) and NAB-Sure (Fig. 7B). We then determined the correlation of each with PRNT ($NT_{50}$). As shown in Fig. 7, there was a significant correlation between RBD MFI values and PRNT $NT_{50}$ ($r_s > 0.8$; $P < 0.0001$). The NAB-Sure proved to be equally robust ($r_s > 0.8$; $P < 0.0001$). We conclude that SARS-COV-2 neutralizing titers can be estimated in DBS eluates with high degree of confidence.

## DISCUSSION

As the World Health Organization (WHO) and other public health agencies integrate long-term COVID-19 disease management strategies into the broader landscape of infectious diseases, DBS have the potential to serve as a centralized biospecimen type from which to estimate levels of natural and vaccine-induced immunity at population scales (12). Indeed, DBS-based seroprevalence and neutralization assays have been established and deployed to monitor vaccine-induced immunity for polio (29), human papillomavirus (HPV) (30), and several bacterial pathogens and parasites (31). However, with SARS-CoV-2 seroprevalence nearing universality as a result of natural infections/reinfections alongside widespread vaccination, the future utility of DBS to COVID-19 will depend on the development of robust methods to assess functional antibody titers alongside established binding antibody unit (BAU) determinations (14, 32).

With this broad objective in mind, we investigated the compatibility of DBS eluates with different SARS-CoV-2 binding (8-plex MIA) and surrogate neutralization assays (iACE2, RVP, and NAB-Sure) and compared the results to PRNT values ($NT_{50}$ and % neutralization) obtained with paired serum samples in a clinically validated assay. We found that the 8-plex MIA, iACE2, and NAB-Sure assays were not only compatible with DBS eluates, but output values generally correlated with PRNT values. In particular, the MIA and NAB-Sure assay strongly correlated with $PRNT_{50}$ values using a relatively large collection of whole blood-derived DBS from known vaccinated and infected individuals.

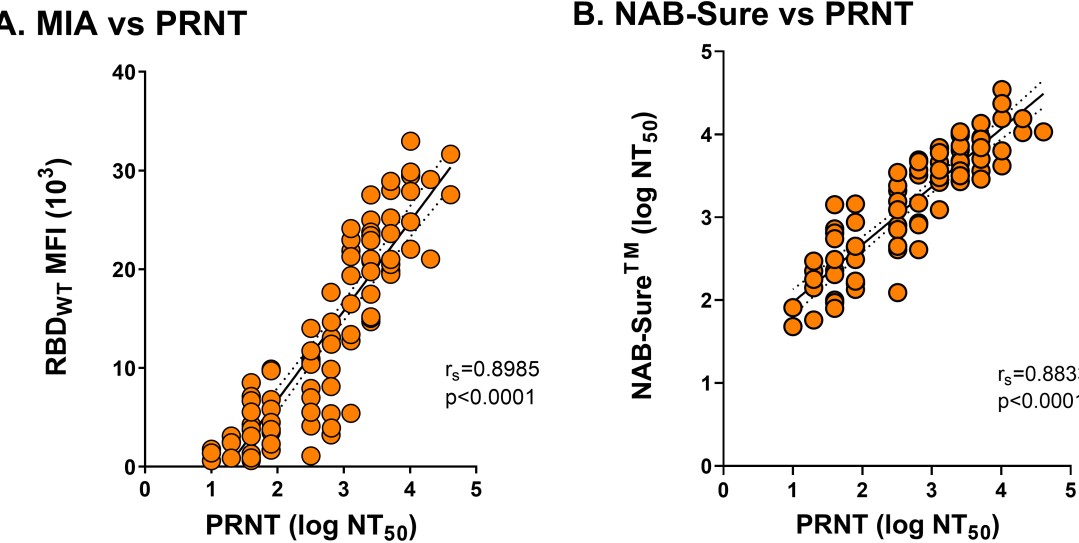

**FIG 7** SARS-CoV-2 neutralizing titers in cDBS eluates from venous whole-blood cDBS. (A) Spearman's correlation of 8-plex MIA $RBD_{WT}$ from venous blood DBS sample eluates ($n = 86$) compared to PRNT $NT_{50}$ paired serum. (**B**) Spearman's correlation of NAB-Sure $NT_{50}(log_{10})$ from venous blood DBS sample eluates ($n = 83$) compared to PRNT $NT_{50}$ ($log_{10}$) paired serum.

The 8-plex MIA and NAB-Sure assay are each amenable to high-throughput testing. Thus, we conclude that DBS are compatible with multiple SARS-CoV-2 surrogate neutralizing assays that have the potential to be applied in population-based surveillance efforts.

The use of DBS as a biospecimen type for estimating SARS-CoV-2 neutralizing antibodies was pioneered by others. Itell and colleagues were the first to report the use of DBS to access functional (neutralizing) antibodies from COVID-19 convalescents and SARS-CoV-2 vaccinees (16). In that report, a cell-based SARS-CoV-2 pseudovirus assay proved to be a reliable measure of neutralizing titers with both mock DBS (venous blood spotted on Whatman filter paper) and self-collected fingerprick DBS from vaccinated individuals. In a larger-scale study, Roper and colleagues reported success "estimating" SARS-CoV-2 pseudovirus neutralizing titers from fingerprick DBS eluates but noted that multiple samples of the 12 tested were eliminated due to values below the limit of detection (33). In that study, fingerprick DBS eluates were not reported at dilutions less than 256, suggesting to us the possibility of interference at low dilutions, which is consistent with our observations using commercially available reporter virus particles. As a substitute for cell-based assays, McDade and colleagues reported the compatibility of DBS eluates with an ACE2-RBD inhibition assay on the Mesoscale Diagnostics platform (17). We had similar success using an in-house iACE2 Luminex-based assay.

The NAB-Sure assay however has the advantage over the pseudovirus and iACE2 assays, in that it was ultrasensitive and potentially amenable to scale-up and automation. As NAB-Sure is a commercial kit, there were no in-house costs associated with reagent quality assurance, making the kit comparable (cost-wise) with most pseudovirus assays. In our hands, NAB-Sure proved robust with DBS eluates in which SARS-CoV-2 levels were on the low end and where cell-based pseudovirus assays proved unreliable due to limited sensitivity.

The use of DBS for infectious disease serology and functional antibody testing is not without intrinsic challenges, including variability in blood collection by fingerpick, uniformity of antibody recovery in eluates, and limited sample volumes (12). Nonetheless, the benefits of DBS as a biospecimen type for surveillance of hard-to-reach communities are well-recognized. For example, in work spearheaded by the CDC, DBS from ~1,000 children in Bangladesh were sampled for neutralizing antibodies to polio types 1, 2, and 3 using a modified poliovirus microneutralization assay (29). In the case of polio, a neutralizing antibody titer of 1:8 correlated with protection and was

used as the threshold definition of seropositivity. Similarly, in a study examining natural versus vaccine-induced antibody responses to human papillomavirus (HPV), neutralizing activity was detected in 2-mm DBS punches eluted into 0.2 mL saline (30). We are also investigating the congruence between SARS-CoV-2 specific antibody titers in oral fluids and DBS, the two biospecimen types might potentially be used interchangeably as a means to survey large cohorts for immunity and vaccine responsiveness (34).

Several limitations of our study should be noted. First and foremost, our conclusions are based on results from contrived and EDTA whole blood-derived DBS, not from DBS samples collected as part of a clinical or population-scale study. However, community-wide studies using self-collected DBS are ongoing in collaboration with C4R (8) as well as a network provider in Northern California . Second, our study is limited to the use of ancestral SARS-CoV-2 virus and WA1-based reagents from the earliest phase of the COVID-19 pandemic. In order to be relevant, it is imperative that we update the MIA, iACE, and NAB-Sure assays reflect circulating SARS-CoV-2 variants. To this end, we are integrating different RBD variants into our iACE2 assay and have adopted NAB-Sure kits based on SARS-CoV-2 Omicron variants and are testing them with self-collected DBS samples.

## MATERIALS AND METHODS

### Commerical human serum panels

The three panels of human sera used in this study (Panels D, E, and H) were obtained from Access Biologicals, LLC, (Vista, CA). Panel D consists of sera from 30 SARS-CoV-2 PCR-positive donors. Positive test results were reported in the initial wave of infections between 3/15/2020 and 4/25/2020 in the United States; and samples were drawn between 6/3/2020 and 6/4/2020, indicating that all donors were infected with the Washington variant of SARS-CoV-2. The panel consists of 21 female and nine male donors, ages ranging from 25 to 66 years, and all donors were symptomatic. Panel E consists of 87 pre-COVID-19 pandemic SARS-CoV-2 sero-negative donors (73 females and 14 males between the ages of 18 and 64 years) collected between 3/24/2017 and 11/9/2018. Panel H consists of serum samples from participants vaccinated with the Moderna COVID-19 (mRNA 1273) vaccine between December 2020 and March 2021. Samples used in this study were collected 12–17 days after their second vaccine dose. Upon receipt, Panels D, E, and H serum samples were stored at −80°C. For the MIA 8-plex validation, we used a commercially available "seroconversion panel" consisting of plasma samples from a single COVID-19-positive donor (Access Biologicals, LLC). The samples were serially collected between March 2020 and May 2020, and infection was confirmed by Access Biologicals using a series of IgG, IgM, and total Ig tests (Gold Standard Diagnostics, DiaSorin, and VITROS) (35).

### Dried blood spots

Contrived DBS (cDBS) were produced by mixing an equal volume of serum with washed human blood cells (HBC) and then spotted (50 µL) onto Whatman 903 filter paper (SigmaAldrich, St. Louis, MO). HBCs were collected by centrifugation (2200 RCF for 8 minutes) from commercially obtained fresh human whole blood [Type O EDTA ] (ZenBio, Durham, NC). Spots were air-dried at room temperature for 4 hours before filter paper was packaged in semipermeable bags with desiccant and stored at −20°C. For venous whole blood DBS, 50 µL of freshly drawn venous EDTA whole blood was spotted onto Whatman 903 filter paper and stored as described above. Matched serum samples derived from a second whole blood tube were collected for PRNT determination. A total of 86 venous DBS samples were tested in the study.

## Plaque reduction neutralization titer (PRNT)

PRNT was determined as described (19, 22, 23). Briefly, twofold serially dilutions of serum were mixed with an equal volume of 150–200 plaque forming units (PFUs) of SARS-CoV-2 (isolate USA-WA1/2020; BEI Resources, NR-52281) and incubated for 1 hour at 37°C, 5% $CO_2$. Serum samples were not heat-inactivated as we have determined in an independent analysis that this step could be eliminated without any loss in sensitivity in $NT_{50}$ determinations (23). The virus:serum mixture (100 µL) was applied to VeroE6 cells grown to 95%–100% confluency in six-well plates, incubated for 1 hour at 37°C, 5% $CO_2$, and then overlaid with complete EMEM (Eagle's Minimal Essential Medium, 2% heat-inactivated FBS, 100 µg/mL penicillin G, and 100 U/mL streptomycin) with 0.6% agar. Two days post-infection, a second agar overlay containing 0.2% neutral red in complete EMEM was applied. Plaque determinations were conducted 1–2 days later. Neutralizing titers were defined as the inverse of the highest dilution of serum providing 50% ($PRNT_{50}$) or 90% ($PRNT_{90}$) viral plaque reduction relative to a virus-only control. Calculated $NT_{50}$ values were log-transformed for graphing purposes.

## Microsphere immunoassays (MIA)

Magplex-C microspheres (Luminex Corp) were coupled with SARS-CoV-2 antigens (Table S1), including three different nucleocapsid antigens (N, N-NA, and NHT) and four spike antigens (RBD, spike S1 [S1], full-length spike [FLS], and full-length trimer [FLT]), as described (21). Also included in the panel was an anti-human IgG3 internal control bead set (Table S1). Coupling concentrations (per $1 \times 10^6$ beads) were as follows: 5 µg for N, N-NA, NHT, S1, and FLT; 2.5 µg for FLS and RBD; and 0.5 µg for the mouse anti-human IgG3 internal control.

A 3-mm DBS punch from each sample was combined with 250 µL elution buffer (Tris-buffered saline, 1% casein blocker; Bio-Rad Laboratories) in a 96-well plate. The DBS punches were eluted for 1 hour at room temperature. Antigen-coated beads (diluted to 1,250 beads/bead set/well with assay buffer [PBS, 2% BSA, pH 7.4]) and 25 µL DBS eluate were combined in a 384-well plate and incubated at 37°C for 30 minutes while shaking (300 rpm) in the dark. Plates were washed for two cycles using the BioTek 405TS microplate washer and 50 µL wash buffer (PBS, 2% BSA, 0.02% Tween-20, 0.05% sodium azide, pH 7.4). Phycoerythrin-tagged goat anti-human IgG (50 µl) was added to each well. Plates were incubated at 37°C for 30 minutes while shaking in the dark and then washed as described above. Antibody–bead complexes were resuspended with 90 µL xMap sheath fluid (Luminex), plates were incubated for 1 minute at room temperature while shaking, and 70 µL from each well was read on a FlexMAP 3D instrument (Luminex Corp). Results are presented in median fluorescence intensity (MFI). Positive, negative, and background controls were included with every plate.

## Reporter virus particle (RVP) neutralization assays

Pseudo-typed lentiviral reporter virus particles (RVP) tagged with Renilla luciferase (Luc) and 293T-hsACE2 cells were obtained from Integral Molecular, Inc. (Philadelphia, PA). Human serum and paired cDBS eluates were diluted 1:50 in DMEM containing 10 mM HEPES [pH 7.4], 10% fetal bovine serum, 0.05% sodium azide, and 100 U/mL penicillin and 0.1 mg/mL streptomycin (Sigma-Aldrich). Paired sample types were serially diluted in triplicate. DBS eluates diluted less than 1:50 (e.g., 1:10 and 1:25) resulted in a high degree of nonspecific RVP inhibition and were not tested further.

RVPs were thawed at 37°C and diluted as per the manufacturer's specification in the cell medium. Equal volumes of RVPs (50 µL) and diluted human serum or DBS eluate were combined and incubated for 1 hour at 37°C with 5% $CO_2$. Positive and negative control wells were included on each assay plate and consisted of RVPs only (100% infectivity control) or no RVPs (0% infectivity control). During RVP–serum incubation, 80%–90% confluent 293T-hsACE2 cells were harvested, and cell density was determined by using a hemocytometer. Cells were diluted to a density of $2 \times 10^5$ cells/mL in cell

culture media, and 100 µL was added to each well of the RVP +specimen plate and mixed with gentle pipetting. Plates were incubated at 37°C with 5% $CO_2$ for 72 hours. Quantification of SARS-CoV-2 RVP infection was determined using the *Renilla*-Glo luciferase assay system (Promega Corp., Madison, WI). Relative light units (RLU) were detected on a SpectroMax iD3 (Molecular Devices, LLC, San Jose, CA) and analyzed with SoftMax Pro 7.0 software. A Z' value was calculated for each plate for assay accuracy and comparability; plates with a Z'<0.5 were repeated. The RLU mean and standard deviation (SD) of 100% infected wells were determined, and a minimum limit of detection was calculated for each plate as = $(Mean- (3 \times SD))_{100\%Infected}$. Samples that failed to reach the LOD cutoff value at the most concentrated dilution were classified as non-neutralizing (<LOD) and assigned a value of 0 for statistical purposes. For each plate, the RLU mean value for 0% infected wells was subtracted from the mean of 100% infected wells, resulting in a normalized infection value (NIV); % normalized RVP neutralization was calculated as = $100-((S_{AVE}/NIV) \times 100)$, where $S_{AVE}$ = the average of duplicate sample RLU values.

## Human ACE2 inhibition (iACE2) assay

Recombinant soluble human ACE2 was generated by the Wadsworth Center's Protein Expression core facility by transfection of pcDNA3-sACE2(WT)–8his into ExpiCHO-S cells using ExpiFectamine (Thermo Fisher Scientific). Plasmid pcDNA3-sACE2(WT)–8his was a gift from Erik Procko (Addgene plasmid # 149268 ; http://n2t.net/addgene:149268 ; RRID:Addgene_149268) and is described herein (36). The soluble hACE2 was affinity-purified on 2 × 1 mL HisTrap Excel (Cytiva) columns connected in series. ACE2 was eluted with a linear gradient of 20 mM Tris-HCl, 500 mM NaCl, and 500 mM imidazole pH 8.0 over 20 column volumes. Fractions containing ACE2, as assessed by SDS-PAGE, were pooled and dialyzed twice against PBS [pH 7.4]. The protein concentration was assessed by Bradford assay with bovine albumin as a standard curve. ACE2 was biotinylated using EZ-Link NHS Biotin (Thermo Fisher Scientific).

Magplex-C microspheres (Luminex) from separate bead regions were coupled with RBD and FLT from SARS-CoV-2 isolate USA-WA1/2020 ("wild type") and resuspended to 2,500 beads/region/well. Diluted beads were transferred to wells of a non-binding 96-well plate; 50 µL of serum or DBS eluate (1:138) was then added to each well and incubated for 1 hour RT with agitation in the dark. Plates were washed twice manually with wash buffer (190 µL), then biotinylated-hACE2 (4 µg/mL) was added to each well, and plates were incubated for an additional 45 min at RT, with shaking in the dark. Plates were washed, probed with streptavidin-PE (1 µg/mL), and incubated at RT for 30 minutes. The plates were washed to remove unbound streptavidin-PE and then analyzed using FlexMap3D. The MFI for ACE2 binding (in the absence of antibody) was derived from a single column of wells containing diluted beads. Inhibition of ACE2 (iACE2) binding is calculated as follows: % iACE2 = 100-[Raw value/(Average of wells with specific VoC beads and no serum or DBS eluate) x 100], per sample well.

## NAB-Sure

The NAB-Sure SARS-CoV-2 Neutralizing Antibody Test (Spear Bio, MA, USA) was performed according to the manufacturer's instructions (www.spear.bio/nabsure). NAB-Sure is a competitive binding assay that measures the ability of antibodies in DBS samples to block SARS-CoV-2 Spike Protein (S1) from binding to recombinant hACE2 in solution. The assay was evaluated using eluates ($n = 11$) from SARS-CoV-2-infected individuals with a range of spike-specific titers provided by SpearBio. Contrived Panel D and H DBS punches (6 mm) were incubated in NAB-Sure elution buffer at 37°C for 1 hour. Afterward, eluates (15 µL) were processed according to the manufacturer's instructions. The test was read using a 7500 Fast Real-Time PCR System (Thermo Fisher Scientific). Cycle threshold (CT) values were converted to $NT_{50}$ values relative to the negative control sample that is devoid of SARS-CoV-2 neutralizing antibodies. The cycle difference between the sample CT value and the negative control CT value was calculated.

Inhibition % was calculated as = $1-1/2^x$, where x = cycle difference. An interpolated $NT_{50}$ value was determined using GraphPad Prism software. To account for the dilution factor, the final $NT_{50}$ value = y*50, where y = interpolated $NT_{50}$ value. Calculated $NT_{50}$ values were log-transformed for graphing purposes.

## Statistical analysis

Statistical analyses were performed using Prism 9 for Windows (GraphPad Software, Boston, MA). Normality was assessed using the Shapiro–Wilk test, and correlations were assessed by either the Pearson correlation coefficient or nonparametric Spearman correlation based on normality results. An unpaired, two-tailed Welch's t-test was performed to determine significance of differences in ACE2 inhibition between groups. Statistical significance of differences in RVP dilutions was assessed using a two-way ANOVA. In all cases, the following apply: *$P \leq 0.05$, **$P \leq 0.01$, ***$P \leq 0.001$, and ****$P < 0.0001$.

## ACKNOWLEDGMENTS

We thank the Wadsworth Center's Protein expression core facility for expressing human ACE2 and the Cell and tissue culture core facility for Vero cell growth media. We gratefully acknowledge Mohammad Uddin for assisting with elution optimization and Bryanna Freitas for assisting with selection of the internal control for MIA. We thank SpearBio, Inc (Woburn, MA) for providing matched serum and DBS samples as well as assistance in optimizing NAB-Sure assay at the Wadsworth Center. The SARS-CoV-2 reagents provided by MassBiologics (Boston, MA) were generated under a Project Award Agreement to NJM from the National Institute for Innovation in Manufacturing Biopharmaceuticals (NIIMBL), United States and financial assistance award 70NANB20H037 from the US Department of Commerce, National Institute of Standards and Technology.

Research reported in this publication was supported by the National Cancer Institute of the National Institutes of Health under Award Number U01CA260508. The content is solely the responsibility of the authors and does not necessarily represent the official views of the National Institutes of Health.

## AUTHOR AFFILIATION

[1]Division of Infectious Disease, Wadsworth Center, New York State Department of Health, Albany, New York, USA

## AUTHOR ORCIDs

William T. Lee https://orcid.org/0000-0003-2883-0391
Monica M. Parker https://orcid.org/0000-0001-7806-968X
Linda M. Styer http://orcid.org/0000-0001-8137-0318
Nicholas J. Mantis http://orcid.org/0000-0002-5083-8640

## FUNDING

| Funder | Grant(s) | Author(s) |
| --- | --- | --- |
| HHS \| NIH \| National Cancer Institute (NCI) | U01CA260508 | Nicholas J. Mantis |
| DOC \| NIST \| National Institute for Innovation in Manufacturing Biopharmaceuticals (NIIMBL) | COVID19-1.01 | Nicholas J. Mantis |

## AUTHOR CONTRIBUTIONS

Katherine Berman, Investigation, Methodology, Writing – original draft, Writing – review and editing | Greta Van Slyke, Investigation | Hayley Novak, Investigation | Jean M. Rock, Formal analysis, Investigation, Methodology, Validation, Visualization, Writing – original

draft, Writing – review and editing | Rachel Bievenue, Investigation, Methodology, Validation, Visualization, Writing – original draft | Amanda K. Damjanovic, Data curation, Formal analysis, Writing – original draft | Kate L. DeRosa, Data curation, Investigation, Methodology | Gianna Mirabile, Investigation | Kara Phipps, Investigation | Jessica Machowski, Investigation | Sean Bialosuknia, Investigation | Roxie C. Giradin, Formal analysis, Investigation, Methodology | Alan P. Dupuis, Investigation | Anne F. Payne, Investigation | Kathleen A. McDonough, Formal analysis, Investigation, Methodology, Project administration, Supervision, Validation | Monica M. Parker, Conceptualization, Funding acquisition, Investigation, Supervision | Linda M. Styer, Conceptualization, Data curation, Formal analysis, Investigation, Methodology, Project administration, Supervision, Validation, Visualization, Writing – original draft, Writing – review and editing | Nicholas J. Mantis, Conceptualization, Funding acquisition, Project administration, Supervision, Visualization, Writing – original draft, Writing – review and editing.

## ADDITIONAL FILES

The following material is available online.

### Supplemental Material

**Supplemental material (Spectrum00846-24-s0001.pdf).** Tables S1 to S4; Fig. S1 to S4.

### Open Peer Review

**PEER REVIEW HISTORY (review-history.pdf).** An accounting of the reviewer comments and feedback.

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
