## [Reviewer comments · Microbiology Spectrum]

Microbiology Spectrum

Quantitating SARS-CoV-2 Neutralizing Antibodies from Human Dried Blood Spots

Katherine Berman, Greta Van Slyke, Hayley Novak, Jean Rock, Rachel Bievenue, Amanda Damjanovic, Kate DeRosa, Gianna Mirabile, Kara Phipps, Jessica Machowski, Sean Bialosuknia, Roxie Girardin, Alan Dupuis, Anne Payne, William Lee, Kathleen McDonough, Monica Parker, Linda Styer, and Nicholas Mantis

Corresponding Author(s): Nicholas Mantis, Wadsworth Center, New York State Department of Health

Review Timeline:

Submission Date:	April 2, 2024
Editorial Decision:	April 25, 2024
Revision Received:	August 24, 2024
Accepted:	August 30, 2024

Editor: William Lainhart

Reviewer(s): Disclosure of reviewer identity is with reference to reviewer comments included in decision letter(s). The following individuals involved in review of your submission have agreed to reveal their identity: Elias A. Rahal (Reviewer #2)

Transaction Report:

DOI: <https://doi.org/10.1128/spectrum.00846-24>

Re: Spectrum00846-24 (Quantitating SARS-CoV-2 Neutralizing Antibodies from Human Dried Blood Spots)

Dear Dr. Nicholas J. Mantis:

Thank you for the privilege of reviewing your work. Below you will find my comments, instructions from the Spectrum editorial office, and the reviewer comments.

Revision Guidelines

Sincerely,
William Lainhart
Editor
Microbiology Spectrum

Reviewer #1 (Comments for the Author):

See attached document.

Reviewer #2 (Comments for the Author):

The paper by Berman et al. examines the utility of using eluates from Dried Blood Spots (DBS) to assess SARS-CoV-2 neutralizing antibodies. For this purpose they compare data from a SARS-CoV-2 plaque reduction neutralization titer (PRNT)

assay conducted on subject serum panels to outputs from multiple surrogate assays.

The paper is well written and the data may serve as preliminary data for assessment of the various surrogate assays tested. The use of DBS to assess neutralizing antibody titers for SARS-CoV-2 is not a novel concept and several groups have tested this. The main difference here is the number of surrogate assays examined.

My main concerns are:

1-The authors argue that the use of DBS to assess neutralizing antibody titers may still be relevant (despite the globally high prevalence of SARS-CoV-2 seropositivity)in particular difficult-to-assess cohorts vulnerable to the emergence of new variants. I believe that it may only be relevant to assess the presence of neutralizing antibodies in populations where these variants are rather highly different from circulating ones. In this case, the surrogate assays assessed will likely not be of use based on their reliance on well-defined antigens rather than ones that may emerge.

2-I find the number of samples assessed to be rather small to derive definitive conclusions. If the merit of the paper is purely technical, then at least one of the promising surrogate assays should be thoroughly assessed using a higher number of samples. This technique would then have to be assessed for its accuracy and precision to derive any meaningful conclusions.

3-It is not quite clear whether the serum samples tested were re-tested by PRNT at the time of contrived DBS preparation or if the authors are relying on PRNT conducted at the time of collection. My concern here is that with the passage of time, older samples may no longer be reliable unless re-assed by PRNT.

4-Was PRNT tested on the serum treated with the washed RBCs prior to drying? My concern is that incubation with the RBCs may affect PRNT.

Editor comments:

Line 214 - Genbank number missing

I question the use of the word "validated"/"validation" as used in this manuscript - especially Lines 244 and 475 (both description of the NAB-Sure product). I understand that there is discussion of 11 specimens with serial dilution and each dilution tested multiple times, but that does not constitute a validation - especially for a RUO product. Please rephrase.

Panel H - this panel is derived from donors who received 2 doses of the Moderna vaccine. Which Moderna vaccine? Additionally, do we know whether any of these donors had also been infected previously with SARS-CoV-2? Some of the results in Figure 6 seem to show two specimens with higher neutralizing antibody titers than the rest of the cohort. Were these the same two donors in all panels?

Response to Reviewers (R2R)

Re: Spectrum00846-24_R1

Reviewer #1

R1.1: Section 2.1 Reagents: this table should be a supplement.

Response: Agreed. As requested, we have moved the SARS-CoV-2 Reagents and Sources to Table S1.

R1.2 Methods section 2.2 / 2.4 / 2.6: Were serum and DBS eluates heat inactivated before use in PRNT or RVP assays? There is no mention of this in the methods. Heat inactivation is important for cell-based immunoassays such as PRNT and RVP due to interference of complement proteins with cell lines. If this was not done, the PRNT data and pseudovirus data are questionable. To address this, authors should use heat inactivated samples to either validate their existing data repeat cell-based neutralization experiments entirely. Response: The Reviewer's comments is duly noted and not the first time we have had to address this issue. The PRNT assay employed in this study is a clinically validated assay used within the New York State Department of Health. In the early days of the SARS-CoV-2 testing serum samples subjected to heat inactivation would often congeal for unknown reasons but likely related to the biology of severe COVID-19. Others have noted such issues:

Hu X, An T, Situ B, Hu Y, Ou Z, Li Q, He X, Zhang Y, Tian P, Sun D, Rui Y, Wang Q, Ding D, Zheng L. Heat inactivation of serum interferes with the immunoanalysis of antibodies to SARS-CoV-2. *J Clin Lab Anal.* 2020 Sep;34(9):e23411. doi: 10.1002/jcla.23411. Epub 2020 Jun 28. PMID: 32594577; PMCID: PMC7361150.

Pastorino B, Touret F, Gilles M, de Lamballerie X, Charrel RN. Heat Inactivation of Different Types of SARS-CoV-2 Samples: What Protocols for Biosafety, Molecular Detection and Serological Diagnostics? *Viruses.* 2020 Jul 7;12(7):735. doi: 10.3390/v12070735. PMID: 32646015; PMCID: PMC7412566.

As a consequence, the Wadsworth Center eliminated the heat inactivation step based on internal studies demonstrating equal PRNT values without or with heat inactivation (unpublished results). Moreover, the Wadsworth Center's PRNT assay (without heat inactivation) was shown to be concordance with with two other public health laboratories in which serum samples were exchanged and PRNT values compared. Those results were published in *Microbiology Spectrum*:

Valcourt EJ, Manguiat K, Robinson A, Lin YC, Abe KT, Mubareka S, Shigayeva A, Zhong Z, Girardin RC, DuPuis A, Payne A, McDonough K, Wang Z, Gasser R, Laumaea A, Benlarbi M, Richard J, Prévost J, Anand SP, Dimitrova K, Phillipson C, McGeer A, Gingras AC, Liang C, Petric M, Sekirov I, Morshed M, Finzi A, Drebot M, Wood H. Evaluating Humoral Immunity against SARS-CoV-2:

Validation of a Plaque-Reduction Neutralization Test and a Multilaboratory Comparison of Conventional and Surrogate Neutralization Assays. *Microbiol Spectr.* 2021 Dec 22;9(3):e0088621. doi: 10.1128/Spectrum.00886-21. Epub 2021 Nov 17. Erratum in: *Microbiol Spectr.* 2022 Aug 31;10(4):e0055322. doi: 10.1128/spectrum.00553-22. PMID: 34787495; PMCID: PMC8597631.

We have updated the Materials and Methods section to indicate that serum samples tested in PRNT were not heat inactivated and cited the article above.

R1.3 Section 2.5: The authors cannot cite a “manuscript in preparation” as a reference to methods described elsewhere. If the information is not accessible to the reader, it must be described in the paper. At a minimum, the authors need to provide a description of the coupling method and internal control antibody and some validation data. Response: Indeed, the Reviewer’s point is well taken. We have therefore expanded the current manuscript to include the full 8-plex microsphere immunoassay MIA, with relevant validation studies. This information is now captured in Tables S2 and S3, as well as Figures S1 and S2.

R1.4 Section 2.9 Stats: Were data tested for normality? This type of data is often not normally-distributed, especially at low n, in which case the authors should be using non-parametric statistics for analysis. For example, in figure 4 it is clear that the % ACE2 inhibition data is not normally distributed. Response: We thank the Reviewer for this specific comment regarding normality with small sample sizes. As requested, we have re-assessed each data set for normality, as determined using the Shapiro-Wilk test. We have updated the Statistical Analysis methods section accordingly. Data sets for Figures 2, 3B and S4 (E) were had normal distributions, and were therefore analyzed using the Pearson correlation coefficient. Data sets associated with Figures 3C, 4B, 5 (A-B) and S4 (C-D) were not normally distributed and therefore the correlations were re-analyzed using the Spearman’s correlation coefficient. Revised r_s and p-value are now included with those figures.

R1.5 Section 3.1 Generation of DBS Panel. As indicated above, samples should have been heat-inactivated for PRNT. Was this done? Text needs to be more clear about specimen type here. Text describes preparation of eluates and PRNT assays but it’s unclear that the PRNT were done with serum samples. Response: Please see comment to R1.2.

R1.6 Figure 1 should include PRNT curves for serum used to establish benchmark. Response: For the Panel D, we provide NT_{50} values in Table 1, as the contrived DBS samples had extremely low Spike-specific antibody levels. However, in response to this comment, we performed full titration curves for human serum samples associated with an internal SARS-CoV-2 vaccine study and provide NT_{50} values in Figure 7.

R1.7 The authors state 90% recovery of IgG. How was % recovery calculated and what was the variability across the sample set? Figure 1 should include this data.

Response: As requested, we provide Supplemental Information demonstrating our efforts to optimize antibody recovery from DBS (see Figure S1). We have removed the specific reference to “90% recovery...” from the revised manuscript because this related to a spike monoclonal antibody in a contrived DBS, not total IgG or even spike-specific IgG in human serum.

R1.8 Figure 1B is both uninformative and misleading. It suggests n=30 Panel D and n=86 samples were used for analysis but it appears this was only done to measure RBD titers, which isn't even the focus of the manuscript. Where indicated, it appears only 15 Panel D and 7 (?) Panel E samples were used. It is indicated that there were n=30 Panel H samples but only n=15 were used (?). Figure 1B should be removed and all figure legends and results sections need to indicate sample numbers actually used. Response: In response to the Reviewer's comment, we have removed Figure 1B. To clarify sample size for each study, we have updated the Results and figure legends for all figures where relevant to indicate number of samples tested for each experiment.

R1.9 The ‘type’ of samples selected is commendable as it includes those from low antibody-convalescent (Panel D –no vax) and high –antibody vaccinated (Panel H +vax) however they originated from the early days of the pandemic where the circulating strain was closer to the ancestral (Wuhan). Neutralizing antibodies are known to be lower since VOC's appeared and while the authors mention this limitation (line 480), they should be asked whether they had the opportunity to test more recent samples and if so be permitted to mention their findings as a Note: Response: Indeed, we have now examined >10,000 DBS samples collected over the past 36 months from individuals across the country and have initiated side-by-side NAB-Sure testing using kits that assess neutralizing activity against Wuhan and Omicron BA4/5. The results of these studies will be included in forthcoming manuscripts. For now, we have edited the text to note that such studies are ongoing and that the current study is limited to WA1.

R1.10 Section 3.2 MIA. Need to use panel E to define threshold for positive response. As shown in Figure S1, the background is high for Nucleocapsid antigens. Response: As requested, we have updated the Materials and Methods describing the use of the 8-plex MIA and the establishment of thresholds (cutoffs) for each of the antigens.

R1.11 It is unclear why the authors would sum the MFIs for the four Spike antigens. They have shared epitopes and are not distinct antigens. This “summed” analysis should be removed. Response: We have found that given DBS eluates collected from human volunteers do have differential reactivity profiles associated with RBD, S1, full length spike, and trimeric spike, as demonstrated in Table S2, suggesting that certain individuals may have antibodies that are bias towards certain spike-specific epitopes (NTD, quaternary RBD configurations, etc). As we amass larger numbers of samples tested in the 8-plex MIA we are investigating whether the “sum of spike” value does provide greater sensitivity than individual values. As this

article is within the Methods and Protocols type, we would prefer to include the “sum...” values for future citation.

R1.12 Section 3.3 Pseudovirus. Authors conclude that cDBS eluates had insufficient titers to distinguish them from unvaccinated controls, but their troubleshooting approach involved diluting rather than concentrating samples. Their first dilution step is 1:50. Did they try diluting less? If they think the titers are too low, this seems like an obvious thing to try. Response: Yes, we did evaluate less diluted samples, but only to discover that the more concentrated cDBS eluates resulted in increased background to the point that we could not differentiate the negative controls from the positive controls. Efforts to eliminate background due to a variety of factors (including hemoglobin) were explored without improvement of the signal to noise ratio. We found that 1:50 was the first point where negative and positive controls can be differentiated. We have updated the Results and the Materials and Methods to acknowledge these efforts.

R1.13 As mentioned with respect to methods section above, cell-based neutralization assays are very sensitive to complement proteins. Did the authors heat inactivate the serum and DBS eluates? Response: As noted above (R1.2), serum and plasma samples were not routinely heat inactivated prior to PRNT assays. Similarly, DBS eluates were used as a dilution but not heat inactivated either, since we eventually moved to the NAB-Sure platform where heat inactivation was deemed unnecessary to achieve near perfect equivalence with PRNT.

R1.14 Section 3.4 ACE2. Need to use Panel E to define threshold for positive response. It is possible that the dotted line in Figure 4B is supposed to be this threshold, but it’s not clear how this was calculated and it looks like it’s lower than several of the Panel E samples. An analysis of ACE2 inhibition in DBS eluate vs serum (in addition to DBS vs PRNT) should be included here.

Response: Thank you for pointing this out. To clarify, we have revised the figure legend to indicate that the dotted line in panel 4B refers to the average (not a threshold per se) of negative control samples in Panel E (Figure 4A). Some Panel D samples did in fact have lower iACE2 values than the average of Panel E and therefore fall below the dotted line. Additionally, based on the Reviewer’s recommendation a correlation plot comparing DBS eluates vs serum was added to **Figure S4** to demonstrate the relationship between DBS eluates and serum (r_s -value = 0.70; p-value of 0.005), indicating a moderate correlation.

R1.15 Section 3.5 NAB-Sure. The text indicates that 7 samples from Panel E were run on the NAB-Sure test but the data are not indicated in Figure 5. Need an analysis to show that the NAB-sure test is able to resolve true neutralizing Ab+ DBS eluates over the background present in negative (i.e. Panel E) samples. As with other assays, need to use Panel E to define a threshold for positive response. Response: Thank you for pointing this out. Although we agree it would have been optimal to use Panel E as a threshold for a positive response here, we were unable to make that analysis with the NAB-Sure assay because The Panel E samples

(pre-2019) had negligible neutralizing activity. Specifically, at the lowest dilution factor (1:2), we the average % inhibition for the Panel E samples was 25%. Due to this, it was not feasible to calculate an NT50 value with such low titers and therefore could not be used as a comparable threshold for Panel D. However, there are negative controls that are run with every plate, as well as QC markers. The addition of Figure 7 also validates the NAB-Sure assay.

R1.16 Figure 5. The inclusion of separate graphs for 3 replicate runs of the assay is redundant. The description of replicate CVs is already described in the text so there is no need to include 3 (almost identical) figures. The three replicates should be averaged and correlated to PRNT once, in a single graph. An additional graph of Panel E vs Panel D (e.g. similar to Fig 4A) should be added.

Response: The Reviewer's critique is acknowledged. We have revised Figure 5 accordingly.

R1.17 Section 3.6 Figure 6B: x-axis should not be labeled "Diasorin liaison". That is an instrument and has not been described anywhere in the manuscript. Relabel x-axis with actual readout of the assay (was it Spike IgG?).

Response: Agreed. We have removed this panel from the figure.

R1.18 Figure 6E,F: data here are represented as NT50 whereas RVP analysis in Figure 3 was shown as % inhibition, so it is impossible to compare results between the different sample sets. What is the NT50 threshold of this assay based on Panel E? Are Panel H samples above the threshold?

Response: As requested, we have revised Figure 6 to show direct comparisons between RVP neutralization, iACE2 and NAB-Sure PRNT for panels D, E and H so that the assays are directly comparable.

R1.19 1 Can the authors explain why the MIA assay was evaluated against Panel D (convalescent –low antibody) but not against Panel H (+vax-high antibody) samples?

Response: We apologize for the oversight. We have included the MIA for Panel H in the Figure 6A.

R1.20 While they make reference to it (lines 94, 455) , why did the authors not consider including the ACE 2 assay from Mesoscale Diagnostics ? Although the authors demonstrated very good performance of their own assay on both low antibody (Panel D) and high antibody (Panel H) it remains an in-house assay which may make it challenging for generalizability and/or implementation.

Response: We had originally sought to do exactly that. However, we do not have access to MSD instrument.

R1.21. Incompatibility of DBS with cell-based neutralization assays may not be accurate if samples were not heat-inactivated.

Response: We have modified the Discussion to include more detailed discussion of study limitations.

Reviewer #2

The paper by Berman et al. examines the utility of using eluates from Dried Blood Spots (DBS) to assess SARS-CoV-2 neutralizing antibodies. For this purpose they compare data from a SARS-CoV-2 plaque reduction neutralization titer (PRNT) assay conducted on subject serum panels to outputs from multiple surrogate assays.

The paper is well written and the data may serve as preliminary data for assessment of the various surrogate assays tested. The use of DBS to assess neutralizing antibody titers for SARS-CoV-2 is not a novel concept and several groups have tested this. The main difference here is the number of surrogate assays examined.

R2.1 The authors argue that the use of DBS to assess neutralizing antibody titers may still be relevant (despite the globally high prevalence of SARS-CoV-2 seropositivity) in particular difficult-to-assess cohorts vulnerable to the emergence of new variants. I believe that it may only be relevant to assess the presence of neutralizing antibodies in populations where these variants are rather highly different from circulating ones. In this case, the surrogate assays assessed will likely not be of use based on their reliance on well-defined antigens rather than ones that may emerge. Response: The Reviewer makes an excellent point. In fact, since submission of this manuscript, we have integrated the most recent Omicron VOCs into the MIA and the iACE2 assay. We are also running Omicron BA4/5 NAB-Sure assays alongside the original Wuhan assay presented in this study. These ongoing studies are included as a separate manuscript (K. Berman, G. Mirabile, and N. Mantis, *manuscript in preparation*). We have updated the Discussion to reflect this point.

R2.2-I find the number of samples assessed to be rather small to derive definitive conclusions. If the merit of the paper is purely technical, then at least one of the promising surrogate assays should be thoroughly assessed using a higher number of samples. This technique would then have to be assessed for its accuracy and precision to derive any meaningful conclusions. Response: In light of Reviewer 2's comments, we have now added an additional 86 whole blood-derived DBS samples from a study conducted at the Wadsworth Center where PRNT values were completed. The matched serum and DBS were subjected to the 8-plex MIA and NAB-Sure testing and results are now provided as Figure 7 showing very high degrees of concordance.

R2.3-It is not quite clear whether the serum samples tested were re-tested by PRNT at the time of contrived DBS preparation or if the authors are relying on PRNT conducted at the time of collection. My concern here is that with the passage of time, older samples may no longer be reliable unless re-assessed by PRNT.

Response: The PRNT values were derived from serum that was frozen immediately following collection, while the NAB-Sure values were derived from DBS cards stored at -20°C. The addition of Figure 7 showing high degree of concordance between paired serum and DBS samples from 86 volunteers attests to sample quality.

R2.4-Was PRNT tested on the serum treated with the washed RBCs prior to drying? My concern is that incubation with the RBCs may affect PRNT.

Response: PRNT values were only obtained with serum, not with reconstituted whole blood or DBS samples. In fact, we agree that RBCs may represent a major confounding factor in the PRNT assays with DBS eluates.

Re: Spectrum00846-24R1 (Quantitating SARS-CoV-2 Neutralizing Antibodies from Human Dried Blood Spots)

Dear Dr. Nicholas J. Mantis:

Your manuscript has been accepted, and I am forwarding it to the ASM production staff for publication. Your paper will first be checked to make sure all elements meet the technical requirements. ASM staff will contact you if anything needs to be revised before copyediting and production can begin. Otherwise, you will be notified when your proofs are ready to be viewed.

Sincerely,
William Lainhart
Editor
Microbiology Spectrum

Reviewer #1 (Comments for the Author):

it is clear that the submitters have read and in many cases made the necessary changes and/or justified why they didn't. the revised version is much cleaner and stronger

Reviewer #2 (Comments for the Author):

The authors responses and modifications to the manuscript satisfy my previous concerns and queries.